# Exploring the Willingness to Accept SARS-CoV-2 Vaccine in a University Population in Southern Italy, September to November 2020

**DOI:** 10.3390/vaccines9030275

**Published:** 2021-03-18

**Authors:** Gabriella Di Giuseppe, Concetta Paola Pelullo, Giorgia Della Polla, Maria Pavia, Italo Francesco Angelillo

**Affiliations:** 1Department of Experimental Medicine, University of Campania “Luigi Vanvitelli”, Via L. Armanni 5, 80138 Naples, Italy; gabriella.digiuseppe@unicampania.it (G.D.G.); concettapaola.pelullo@unicampania.it (C.P.P.); maria.pavia@unicampania.it (M.P.); 2Health Direction, Teaching Hospital of the University of Campania “Luigi Vanvitelli”, Via S. Maria di Costantinopoli 104, 80138 Naples, Italy; giorgia.dellapolla@unicampania.it

**Keywords:** attitudes, COVID-19, university population, Italy, vaccination, willingness

## Abstract

Understanding whether members of the university population are willing to receive a future vaccination against COVID-19 and identifying barriers may help public health authorities to develop effective strategies and interventions to contain COVID-19. This cross-sectional study explored the willingness to accept a future SARS-CoV-2 vaccine in a university population in Southern Italy. The perceived risk level of developing COVID-19 was 6.5 and it was significantly higher among females, younger subjects, and those who agreed/strongly agreed that COVID-19 is a severe disease. Only 21.4% of respondents were not worried at all regarding the safety of the vaccine. Males, not being married/cohabitant, being a faculty member, those who perceived a lower risk of developing COVID-19, and those who did not need information regarding the vaccination against COVID-19 were significantly more likely to have no concern at all regarding the safety of the vaccine. The vast majority (84.1%) were willing to receive a future vaccine against COVID-19. Almost coherently with predictors of concern on the safety of the vaccine, being male, not being married/cohabitant, being a faculty member, not being concerned at all that COVID-19 vaccination might not be safe, and agreeing that COVID-19 can have serious health consequences were significant predictors of the willingness to receive the vaccine against COVID-19. A considerable proportion of the population had a positive willingness to receive the future COVID-19 vaccine, although some concerns have been expressed regarding the effectiveness and safety and public health activities seem necessary to achieve the rate that can lead to the protection of the community.

## 1. Introduction

Coronavirus disease 2019 (COVID-19), caused by the severe acute respiratory syndrome coronavirus-2 (SARS-CoV-2), represents a global public health emergency that spread from China in December 2019 to almost all countries around the world. Consequently, on March 11, 2020, the World Health Organization (WHO) declared that COVID-19 was a pandemic disease [1]. The COVID-19 pandemic has now affected more than 115 million people worldwide, of which more than 2.5 million have died as of March 6, 2021 [2]. In Italy, the statistics showed more than three million patients infected with SARS-CoV-2 and 99,271 deaths due to COVID-19 [3]. Considering the extraordinary diffusion of the disease, all countries adopted strategies to reduce the transmission, such as nationwide lockdowns, contact tracing, keeping distance, and individual protective measures, which reduced the burden of COVID-19, though causing a significant economic loss worldwide. However, the pathogen is still continuing to spread with an enormous burden of morbidity and mortality, and this highlights the urgent need for the development of vaccines. Many governments and industries have invested in the development of several efficacious and safe vaccines against COVID-19, and their introduction is the most efficient way to achieve individual- and population-level immunity and to control the global COVID-19 pandemic. In this context, the administration of the vaccine amongst the public at large is crucial. However, when this survey was designed and conducted, no safe and effective vaccine had been licensed yet. Therefore, understanding whether members of the general population are willing to receive a future vaccination against COVID-19 and identifying barriers may help public health authorities to develop effective strategies and interventions to contain COVID-19. Since to the best of our knowledge there is a paucity of published data in Italy, an investigation is imperative. The present cross-sectional study aimed to investigate the willingness regarding a future vaccine against COVID-19 in a university population in Southern Italy and to detect the potential influencing factors to provide evidence for recommendation and exploration of scale-up immunization programs.

## 2. Materials and Methods

### 2.1. Participants

The present survey was conducted as part of a large research project among several groups of individuals. Between September 14 and November 30, 2020, this survey was carried out in the cities of Caserta and Naples, located in the Campania region in the southern part of Italy. Sampling consisted of a non-probability method via the use of a convenient sample of students and employees (permanent or temporary) at the public University of Campania “Luigi Vanvitelli”, who attended the health surveillance center for a voluntary antibody-testing for anti-SARS-CoV-2 immunoglobulin G (IgG) and M (IgM). The only exclusion criterion was for the employees in health-care settings since they were included in a survey among health-care workers [4].

Sample size was estimated with the prevalence of respondents who were willing to receive a future vaccine against COVID-19 set at 50%, a relative precision of 5%, a 95% confidence interval, and a non-response rate of 20%. The final estimated minimum sample size considering the non-response rate was 481.

### 2.2. Procedures

Prior to the enrolment into the study, the participants received an information sheet from the research team with a detailed explanation of the research purpose, the interview procedures, that the participation was entirely voluntary, that they were able to withdraw from the survey at any time if they felt uncomfortable, and that all responses were collected anonymously and confidentially with no identifiable information gathered on the respondents by the research team. Written informed consent was obtained from each participant at the beginning of the interview. All data were collected through a face-to-face interview by a previously trained team in a private office setting to provide a safe place for the participants to share their perceptions without judgment and the influence of others or through a self-administered questionnaire. No incentives were offered for participation in the survey.

### 2.3. Ethical Approval

The study was approved by the Ethics Committee of the Teaching Hospital of the University of Campania “Luigi Vanvitelli”.

### 2.4. Study Instrument

Information was collected through a standardized, anonymous, structured questionnaire. The tool had been piloted before the actual data collection by taking a sample of 50 participants who assessed the content of the questions and their acceptability and comprehensibility, and the results were not included in the survey. The instrument consists of four major parts. In the first part of the questionnaire, the questions addressed individuals’ socio-demographic and general characteristics such as age, gender, marital status, highest level of education achieved, type of employment, workplace, people who had contracted COVID-19 had the participants had contact with, and degree course and study year (for students); individuals’ health conditions were also collected, including chronic health conditions. The second part investigated participants’ likelihood to receive a future COVID-19 vaccine and reasons for vaccination willingness or unwillingness, and rated their agreement on a 5-point Likert scale, with “1” indicating strongly agree, “2” for agree, “3” for uncertain, “4” for disagree and “5” for strongly disagree, for items regarding the perceived importance of the COVID-19 vaccine, perceived vaccine safety, concerns about the vaccine, the perceived severity of COVID-19, and the level of trust in COVID-19 vaccine information. The categories, disagree, strongly disagree, agree and strongly agree, were collapsed into two categories (disagree and agree) for analysis. Risk perception of contracting COVID-19 was evaluated on a 10-point Likert-type scale from 1 to 10, with 1 indicating the lowest perceived risk and 10 indicating the highest. The third part collected information on the participants’ seasonal influenza vaccine behavior in the previous year and their likelihood to receive the vaccine this year and reasons for vaccination willingness or unwillingness. In the final part, participants were asked to identify which information sources (TV, radio, newspapers, Internet, scientific journals, etc.) had been used about COVID-19 vaccination and whether they were interested in receiving additional information. The participants were able to indicate more than 1 answer regarding their sources of information. A copy of the questionnaire is reported as a Appendix A.

### 2.5. Data Analysis

Descriptive statistics were used to summarize participants’ characteristics and answers to all items with absolute and relative frequencies for categorical variables, and mean and standard deviation for continuous ones. Univariate analyses were performed using the chi-square tests of association for categorical variables and t tests for continuous variables to assess relations between the different outcomes of interest and several characteristics. According to Hosmer and Lemeshow, variables with a *p*-value ≤ 0.25 were subsequently entered into the multivariate regression models, and the significant level choices for the inclusion and elimination of the variables in the models were *p*-values of 0.2 and 0.4, respectively [5]. Multivariate stepwise linear and logistic regressions were used to determine the significant predictors of these three following outcomes: perceived risk of developing COVID-19, which was measured with a value ranging from 1 “low” to 10 “high” (Model 1); concern that the future COVID-19 vaccination might not be safe, which was dichotomized into concern (slight/moderate/some/extreme = 0) versus no concern (not at all = 1) (Model 2); and willingness to receive a future vaccine against COVID-19, which was dichotomized as 1 if the answer was “yes” and 0 if it was “no” (Model 3). The following selected independent variables were included in all multivariate linear and logistic regression models: gender (male = 0; female = 1); age, in years (continuous); marital status (unmarried/separated/divorced/widowed = 0; married/cohabitant = 1); education level (high school degree or less = 0; college degree or higher = 1); professional role [faculty members = 1; students = 2; administrative staff = 3; others (technicians, laboratory, security and cleaning staff) = 4]; having at least a chronic medical condition (no = 0; yes = 1); contact with a confirmed COVID-19 case (no = 0; yes = 1); having experienced in the previous ten months at least one symptom compatible with COVID-19 (no = 0; yes = 1); perceived severity of COVID-19 (disagree/strongly disagree/uncertain = 0; agree/strongly agree = 1); use of mass-media and Internet as sources of information about the vaccination against COVID-19 (no = 0; yes = 1); and needing additional information regarding the vaccination against COVID-19 (no = 0; yes = 1). In Models 2 and 3, the variable perceived risk of developing COVID-19 (continuous) was included, and the variable concern that the future COVID-19 vaccination might not be safe (slightly/moderately/somewhat/extremely = 0; not at all = 1) was included in Model 3. Beta coefficient (β) for each variable in the multivariate linear regression model is presented. In the multivariate logistic regression models, odds ratios (ORs) and their 95% confidence intervals (CIs) were used in the measurement of independent associations between the different variables and the outcomes of interest. For all analyses, two-sided *p*-values of 0.05 or less were considered statistically significant. All analyses were conducted using STATA statistical package version 15 [6].

## 3. Results

### 3.1. Socio-Demographic and Professional Characteristics

All 1518 subjects who attended the health surveillance center agreed to be interviewed. The surveyed population’s characteristics are summarized in Table 1. Slightly less than two-thirds were females, the average age was 36, one-third was married/cohabitant, more than half were students, only 19.4% had at least one chronic condition, only 21.3% had had at least one common symptom compatible with COVID-19 in the previous ten months, and almost one-third (32.2%) knew or had contact with at least one confirmed COVID-19 case, mainly relatives.

### 3.2. Attitudes

In regards to attitudes towards the COVID-19 disease and vaccine, the self-reported risk perception of developing the disease, measured on a 10-point Likert-type scale, resulted in a mean value of 6.5, with 3.4% and 9.7% of the respondents who believed the risk of infection to be 1 and 10, respectively. Table 2 presents the results of the multivariate linear and logistic regression models predicting the different outcomes of interest. The multivariate linear regression model predicting the respondents’ risk perception of developing COVID-19 showed that three independent explanatory variables made a unique statistically significant contribution to the model: gender, age, and awareness that COVID-19 might be a severe disease. The risk of developing COVID-19 was perceived to be higher by females, younger subjects, and by those who agreed or strongly agreed that COVID-19 is a severe disease (Model 1).

The vast majority of the respondents agreed (either partially or completely) with the statements that COVID-19 can have serious health consequences (85.2%) and that it is important to receive this vaccination (82.5%). Despite this generally positive attitude, some concerns persisted, since only 21.4% and 13.2% of respondents were not worried at all regarding the safety and efficacy of the vaccine, respectively. The results of the multivariate logistic regression model showed that five independent predictors were significantly associated with no concern at all about the safety of COVID-19 vaccination. Males, not being married or cohabitant, those who perceived a lower risk of developing COVID-19, those who reported they did not need additional information regarding the vaccination against COVID-19, and being a faculty member compared to administrative staff and other professionals were more likely to have no concern at all regarding the safety of the vaccine against COVID-19 (Model 2 in Table 2). The vast majority (84.1%) of the respondents were willing to receive a future vaccine against COVID-19. Overall, respondents considered the effectiveness (28.9%) of the vaccine and the severity of COVID-19 (25.5%) the most important reasons to receive this vaccination, whilst the safety (20.8%) of the vaccine was the third. Concerns about the adverse effects of the vaccine ranked highest (46.1%) among the reasons for refusing this vaccination uptake, followed by issues related to vaccine effectiveness (36.7%). The results of the multivariate logistic regression model revealed that respondents’ statistically significant predictors of the willingness to receive a future vaccine against COVID-19 included gender, marital status, professional role, perceived severity of COVID-19, and vaccine safety issues. Concern regarding the safety of the vaccine and the perceived severity of COVID-19 were the two strongest predictors indicating that respondents who were not concerned at all and those who agreed (either partially or completely) with the statement that COVID-19 can have serious health consequences were 10 times and 3 times more willing to receive the vaccine against COVID-19. Moreover, male gender, not being married or cohabitant, and being a faculty member compared to administrative staff and other professionals were significant predictors of the willingness to receive the vaccination (Model 3 in Table 2).

Finally, half of the sample (51.6%) would be vaccinated against seasonal influenza in the current season and the main reason was that symptoms of COVID-19 and influenza can be very similar (38.3%). Only 11.1% declared to have been vaccinated for influenza in the 2019–2020 season.

### 3.3. Sources of Information

Almost all respondents had heard about COVID-19 vaccination (99.6%) and they could choose multiple sources of information, among which the most common were TV, radio, and newspapers (63.1%). Participants also mentioned seeking information from the Internet (58.1%) and scientific journals (31.1%). However, a total of 68.9% participants acknowledged that they needed additional information about COVID-19 vaccination for clarification.

## 4. Discussion

The main objectives of this study were to investigate the Italian university population’s attitudes to undergo a future vaccination against COVID-19 and the predictors of their willingness to receive the vaccine. The current survey has important implications for health policy makers and providers and for interventions aimed at promoting the COVID-19 vaccination among the general population.

It is important to consider these results within the context of COVID-19 events. At the time of this study, the COVID-19 vaccine was not on the market, and it would have been available for health-care workers and people aged 80 years and older by the end of 2020. The main finding of the study is that the large majority of the surveyed population (84.1%) expressed the intention to undergo the vaccination against COVID-19. This result is remarkable since Italy, in the past few years, has faced a sustained reduction in the recommended vaccination coverage, which yielded to impose childhood vaccination uptake for access to schools [7]. Taking into account the different impact of the pandemic across countries, which may represent an extraordinary driver for the intention to vaccinate, it is interesting to perform comparisons of the willingness to undergo COVID-19 vaccination among countries. Overall, the population fraction that exhibited the intention in this study is among the highest encountered in the up to date literature, with most studies revealing that the willingness to receive the COVID-19 vaccination includes more than 65% of the general population [8,9,10,11], with only few investigations exceeding 80% [12,13,14,15,16]. In particular, the present results showed a higher willingness compared with previous studies conducted in Europe [9], Japan [11] and in the US [8,10], similar to those reported in Australia [15] and China [13,16]. Instead, a higher willingness was found in Indonesia [12] and in China [14], while willingness was about half in another study conducted in China [17].

Despite this very positive attitude, participants raised a number of issues to explain their unwillingness to receive the future vaccine. The two most common reasons or barriers for COVID-19 vaccination refusal were uncertainty about the vaccine’s efficacy and fear of eventual adverse effects. Similar to these findings, previous studies among different groups of individuals showed that the respondents were reluctant to receive this vaccine, presenting concerns about side effects and efficacy or a low perceived risk of contracting the infection [9,12,18]. Consequently, the presence of such barriers constitutes not only an obstacle to accept the vaccine at the individual level, but also indicates an adverse attitude towards the COVID-19 vaccine that can put these subjects at risk of transmitting the disease. It should be argued, however, that at the time of the study, detailed characteristics of the candidate vaccines related, for example, to safety and effectiveness were not yet available. Therefore, attitudes were evaluated on the potential attributes of the vaccines, and it may be hypothesized that even these barriers might be overcome when considering the actual performance of licensed formulations. However, it is well-known that the process of decision-making and the implementation of health-related behaviors, such as vaccinations, are multifactorial, and the large body of literature that has explored factors influencing the willingness to receive the vaccine has demonstrated that hesitancy is widespread, regardless of knowledge on effectiveness or safety of the specific vaccine. Indeed, effectiveness and safety had been repeatedly demonstrated for the unwillingness to receive vaccines [19,20,21,22,23]. Therefore, the results of this study represent an interesting perspective, even for researchers who are involved in the investigation of the evolving willingness to accept COVID-19 vaccines, as long as new data on the experimental and real-world performance become available. Now that initial results of undergoing trials are becoming available, targeted educational interventions to address the fears of people toward a specific vaccine’s potential or perceived harms, as well as its important role for preventing the spread of the disease, should be implemented as a crucial step for a successful prevention strategy against COVID-19. This strategy is also supported by the results on predictors of willingness to receive the COVID-19 vaccine which showed, consistent with previous research [8,9,10,12,13,14,18,24,25,26], that the absence of concern on the safety of the vaccine was a strong determinant of willingness, as well as the awareness of the potential severity of COVID-19. This latter finding has already been analyzed in previous studies, which found that a high perception of benefits of the vaccination, that is, to protect from a severe disease, was a predictor of the intention to be vaccinated [8,13]. The finding that males were more willing to be vaccinated has already been reported [27], although contrasting results showing a higher intention to be vaccinated among women have also been described [28,29].

Another peculiar finding in this survey is that almost all participants (99.6%) had heard of the COVID-19 vaccine. Respondents indicated that they are frequent consumers of online health information via sources outside the health care system, mainly TV, radio, newspapers, and the Internet. However, it should be noted that those who are seeking this information are likely to find thousands of websites and online video platforms and their quality and accuracy need attention. This finding adds to research evidence that indicates that poor quality or inaccurate information from such sources can have negative effects on health behaviors and the uptake of recommended public health interventions. Nonetheless, the findings of the present study also determined that fewer respondents reported hearing about the vaccine from scientific journals. Moreover, it is well known that health-care workers, mainly physicians involved in primary and preventive care, should intensively promote vaccination because their recommendation and information dissemination to patients has a significant impact on vaccination rates [30,31,32,33]. Therefore, health organizations and health-care workers are required to provide, especially in the event of a new disease such as COVID-19 that is globally at the center of attention, communication regarding immunization by coupling their recommendation with clear messages regarding vaccine safety, effectiveness, and benefits.

It is important to note that the results from this survey should be interpreted with the potential following methodological limitations. Firstly, the cross-sectional design used may limit the ability to identify causal relationships between the several independent variables and the different outcomes of interest. Secondly, the generalizability of the results may be limited. The selected population included only the personnel and the students of one university in Southern Italy, and this may not reflect the perceptions and attitudes of the general population in the country as a whole, since the sample is probably composed of more educated subjects compared to the general population. Moreover, the subjects were selected using a non-probability convenience sampling of those who volunteered to participate in a COVID-19 serologic screening program, and they may be probably more interested in the specific topic. Thirdly, participants might want to give socially desirable answers, which could likely result in a possible overestimation of the proportion of those who were willing to receive the vaccine. However, this limitation may have been mitigated by the measures taken to ensure confidentiality and anonymity, and this may have made respondents more likely to answer the questions accurately and honestly. Finally, since at the time of the study COVID-19 vaccines were not yet available, acceptability should be monitored as long as new information on the characteristics of licensed vaccines will become accessible, including the number of doses, the expected duration of immunity, and experimented adverse events, since they may affect willingness and actual vaccine uptake. Despite the limitations described, this study contributes to building an understanding of the general population’s awareness and willingness to receive the future COVID-19 vaccine.

## 5. Conclusions

In summary, the results from this survey suggest that a considerable proportion of the studied population had a positive willingness to receive the future COVID-19 vaccine, although some concerns have been expressed regarding the effectiveness and safety. In light of the findings, public health activities seem necessary to achieve the rate that can lead to the protection of the community.

## Figures and Tables

**Table 1 vaccines-09-00275-t001:** Respondents’ socio-demographic and anamnestic characteristics.

Characteristics	N = 1518	%
Age, years	36 ± 14.2 (18–73) *	
Gender		
Female	923	60.8
Male	595	39.2
Marital status		
Married/cohabitant	504	33.3
Unmarried/widowed/separated/divorced	1009	66.7
Education level		
High school degree or less	804	53
College degree or higher	714	47
Having at least a chronic medical condition		
No	1224	80.6
Yes	294	19.4
Professional role		
Students	794	52.3
Administrative staff	406	26.8
Faculty members	214	14.1
Others (technicians, laboratory, security and cleaning staff)	104	6.8
Contact with a confirmed COVID-19 case		
No	1029	67.8
Yes	489	32.2
Exposure to confirmed SARS-CoV-2 infection ^		
Relatives	156	31.9
Friends	141	28.8
Co-workers	101	20.7
Family members	68	13.9
Household members	23	4.7
Having at least one common symptom compatible with COVID-19 in the last ten months		
No	1195	78.7
Yes	323	21.3

* Mean ± Standard deviation (range). ^ Among those who have had contact with a confirmed COVID-19 case. Number for each item may not add up to total number of study population due to missing values.

**Table 2 vaccines-09-00275-t002:** Multivariate linear and logistic regression analysis results examining the outcomes of interest according to several explanatory variables.

Variable	Coeff.	t	*p*
**Model 1. Perceived risk of developing COVID-19 (Sample size = 1504)**			
*F* (6, 1497) = 26.1, *p* < 0.0001, *R*^2^ = 9.5%, adjusted *R*^2^ = 9.1%			
Younger	−0.02	−3.69	<0.001
Females	0.8	7.27	<0.001
Agree/strongly agree that COVID-19 is a severe disease	1.23	8.34	<0.001
Married/cohabitant	0.25	1.65	0.099
Needing additional information regarding the vaccination against COVID-19	0.14	1.25	0.213
Having had in the previous ten months a common symptom compatible with COVID-19	0.15	1.21	0.226
	**OR**	**95% CI**	***p***
**Model 2. No concern that the future COVID-19 vaccination might not be safe (Sample size = 1493)**			
Log likelihood = −716.2, χ^2^ = 116.64(9 df), *p* < 0.0001			
Males	0.58	0.44–0.76	<0.001
Professional role			
Faculty members	1 *		
Administrative staff	0.39	0.26–0.58	<0.001
Others	0.4	0.22–0.74	0.003
Did not need additional information regarding the vaccination against COVID-19	0.47	0.36–0.61	<0.001
Not being married/cohabitant	0.56	0.37–0.82	0.004
Perceived a lower risk of developing COVID-19	0.93	0.87–0.99	0.029
Having not received information regarding the vaccination against COVID-19 from mass media and the Internet	0.75	0.53–1.06	0.106
Agree/strongly agree that COVID-19 is a severe disease	1.32	0.89–1.96	0.162
Older	1.01	1–1.02	0.164
**Model 3. Willingness to receive a future vaccine against COVID-19 (Sample size = 1501)**			
Log likelihood = −567.85, χ^2^ = 177 (8 df), *p* < 0.0001			
Not concerned at all about the safety of the vaccination	10.4	4.53–23.86	<0.001
Agree/strongly agree that COVID-19 is a severe disease	3.1	2.17–4.42	<0.001
Professional role			
Faculty members	1 *		
Administrative staff	0.48	0.33–0.68	<0.001
Others	0.58	0.34–0.99	0.049
Males	0.67	0.48–0.92	0.015
Not being married/cohabitant	0.69	0.48–0.98	0.037
Perceived to be at a higher risk of developing COVID-19	1.05	0.98–1.13	0.147
High school degree or less	0.84	0.62–1.15	0.288

* Reference category.

## Data Availability

The data presented in this study are available on request from the corresponding author.

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
