# Peer review of "Exploring the Willingness to Accept SARS-CoV-2 Vaccine in a University Population in Southern Italy, September to November 2020"

_vaccines, 2021, doi:10.3390/vaccines9030275_

Round 1

Reviewer 1 Report

This is an interesting article exploring the willingness to accept SARS-CoV-2 vaccine in the general population in Italy, September to November 2020. The procedure applied is well described and results properly presented. Possible limitations of the survey have been taken into consideration in the discussion.

Line 19: were

Line 335: Check the name of the first author

Author Response

Line 19: were

As suggested, we have replaced were with “was” referring to “Only 24.4%”.

Line 335: Check the name of the first author

As suggested, we have corrected the name of the first author.

Reviewer 2 Report

The study by Giuseppe  et al discusses perception about the COVID-19 vaccines in a university in Italy. The authors have surveyed more than 1000 individuals. Unfortunately, the study is not relevant anymore since the survey was done when the data about vaccine efficacy was unknown. I also feel the study does not belong to the Vaccine journal, since it is more suited for a statistical journal or social science/interdisciplinary journal. I have some other comments.

  1. “The two most common reasons or barriers for COVID-19 vaccination 243 refusal as reported by the respondents were uncertainty about the vaccine efficacy and fear of eventual adverse effects” The study has lost relevance since this issue has been addressed with data on vaccine efficacy. So what is the point of this study now ?

2. "The present cross-sectional study aimed to investigate the willing

ness regarding a future vaccine against COVID-19 among the general population” The study is not aimed at general population where individuals would not be as aware as university employees and students.

3. “Between September 14 and November 30, 2020, this survey was

 carried out in the cities of Caserta and Naples located in the Campania region, Southern part of Italy.” The survey was conducted before the vaccine efficacy data for most vaccines were known therefore in absence of any evidence this may serve as another variable in decision making.

4. “Variables with a p-value < 0.25 were subsequently….”How the p value was decided.

5. “professional role (faculty members=1; students=2; administrative staff=3; others=4);” How scores were decided for this group. Also, what is others?

6. “use of mass-media and Internet as sources of information about the vaccination against COVID-19” What are the other source of information ? It seems this would include all source of information on vaccines.

7. Is there a different unmarried male and female groups or vice versa? Such analysis is not done here.

 8. I also think authors should include questionnaires as supplementary file.

9. Though stats is well presented in Table 2, it would be easier to follow for readers of Vaccine if it is presented in figure.

Author Response

1.“The two most common reasons or barriers for COVID-19 vaccination 243 refusal as reported by the respondents were uncertainty about the vaccine efficacy and fear of eventual adverse effects” The study has lost relevance since this issue has been addressed with data on vaccine efficacy. So what is the point of this study now?

In response to this point, it is well-known that the process of decision-making and implementation of health-related behaviors, such as vaccinations is multifactorial and the large body of literature that has explored factors influencing the willingness to uptake vaccinations, has demonstrated that hesitancy is widespread, regardless of knowledge on effectiveness or safety of the specific vaccine, as reported by previous studies that showed the unwillingness to receive also vaccines, whose effectiveness and safety had been repeatedly demonstrated (Morrone 2017, Luz 2017, Napolitano 2017, Bertoldo 2018, Casalino 2018). Indeed, the top three reasons for vaccine hesitancy are related to 1) beliefs, attitudes, motivation about health and prevention, 2) perceived risk/benefit and experiences with vaccines, and 3) communication and media environment, with major issues related to fear of side effects of vaccination and distrust in the vaccines, lack of perceived risk of vaccine-preventable diseases and the influence of anti-vaccination reports in the media (World Health Organization. Report of the SAGE Working Group on Vaccine Hesitancy, 1 October 2014. <http://www.who.int/immunization/sage/meetings/2014/october/1_Report_WORKING_GROUP_vaccine_ hesitancy_final.pdf). Therefore, we are confident that the results of this study represent an interesting perspective, even for those who are interested in the evolving willingness to accept COVID-19 vaccines, as long as new data on the experimental and real-world performance of new COVID-19 vaccines become available. We have now included this issue in the Discussion section.

  1. "The present cross-sectional study aimed to investigate the willingness regarding a future vaccine against COVID-19 among the general population” The study is not aimed at general population where individuals would not be as aware as university employees and students.

As suggested, we have addressed these issues in the Limitations, where we argued about the generalizability of the results, addressing in particular the problems related to the underlying population and the non-probabilistic choice of the sample. Moreover, the title has been changed in “Exploring the willingness to accept SARS-CoV-2 vaccine in a university population in Southern Italy, September to November 2020”.

  1. “Between September 14 and November 30, 2020, this survey was carried out in the cities of Caserta and Naples located in the Campania region, Southern part of Italy.” The survey was conducted before the vaccine efficacy data for most vaccines were known therefore in absence of any evidence this may serve as another variable in decision making.

See answer to question 1

  1. “Variables with a p-value < 0.25 were subsequently….” How the p value was decided.

As suggested, we have clarified in the methods section that the variables selection was performed according to the strategy by Hosmer and Lemeshow (Hosmer DW, Lemeshow S. Applied Logistic Regression. 2nd Edition, New York: Wiley; 2000). In particular, variables that exhibited a <0.25 p-value at the univariate analysis were included in the models.

  1. “professional role (faculty members=1; students=2; administrative staff=3; others=4);” How scores were decided for this group. Also, what is others?

As suggested, we have clarified in the methods and in Table 1 that the “others” category included technicians, laboratory, security and cleaning staff. The number attributed to each variable does not represent a score, since this variable was constructed as a nominal (categorical) one, and the number was assigned in order to generate the dummy variables.

  1. “use of mass-media and Internet as sources of information about the vaccination against COVID-19” What are the other source of information? It seems this would include all source of information on vaccines.

As suggested, we have clarified in the methods section the other sources of information.

  1. Is there a different unmarried male and female groups or vice versa? Such analysis is not done here.

In response to this point, we performed multivariate linear and logistic regression models to investigate the independent role of each of the variables included in the models on the outcomes of interest. Therefore, the independent role of gender (being males vs being female), marital status (being married/cohabitant vs being unmarried), etc. has been explored on the different outcomes (Willingness to receive COVID-19 vaccination, etc.). To avoid misunderstanding we have rephrased the sentences where needed.

  1. I also think authors should include questionnaires as supplementary file.

As suggested, we have included the English version of questionnaire used in this study as Supplementary file.

  1. Though stats is well presented in Table 2, it would be easier to follow for readers of Vaccine if it is presented in figure.

Table 2 presents the results of the multivariate regression models and the presentation represents the standard in the literature and allows comparisons with all other studies exploring the same topic and this presentation has been used in all epidemiological studies conducted by us and also in the last manuscript that has been published by some by some of us on Vaccines (Napolitano et al.. Vaccinations and Chronic Diseases: Knowledge, Attitudes, and Self-Reported Adherence among Patients in Italy. Vaccines 2020, 8, 560) Although, in order to take into account your suggestion we have removed the column regarding the standard error.

Reviewer 3 Report

The article addresses a very important public health problem at this time. It is encouraging that a relatively high proportion of their subjects (84.1%) reported willingness to receive a future Sars-Cov-2 vaccine. Unfortunately, the main issue with the study is that their study population is unlikely to be representative of the "general population in Italy". This was only briefly addressed as the second limitation in the Discussion. This limitation should be thoroughly addressed, starting with the title, as the results are probably not generalizable (this population is made of mostly educated subjects- students, faculty and other from a University, with 99.6% having some degree of knowledge about the vaccine) and all volunteered for antibody-testing against Sars-Cov-2, suggesting an increased level of interest already. As such, all the other results need to be interpreted with caution. 

A few other comments that may represent minuses of the manuscript:

  • the presentation of the results along the 2 models (ie in Table2), as well as their significance is relatively difficult to follow, more clarity is necessary.
  • correlation between models can potentially be discussed
  • the 3rd part from Methods (referring to the behavior about influenza vaccine) was not addressed at all 
  • in the Discussion, would have been good to discuss what other studies investigated the willingness to accept vaccine and maybe address the potential differences between populations
  • More formal problems with the manuscript refer to necessary revision for English, duplicate entire sentence in the Abstract - surprising lack of attention (...Males, not being married...) 

Author Response

The article addresses a very important public health problem at this time. It is encouraging that a relatively high proportion of their subjects (84.1%) reported willingness to receive a future Sars-Cov-2 vaccine. Unfortunately, the main issue with the study is that their study population is unlikely to be representative of the "general population in Italy". This was only briefly addressed as the second limitation in the Discussion. This limitation should be thoroughly addressed, starting with the title, as the results are probably not generalizable (this population is made of mostly educated subjects- students, faculty and other from a University, with 99.6% having some degree of knowledge about the vaccine) and all volunteered for antibody-testing against Sars-Cov-2, suggesting an increased level of interest already. As such, all the other results need to be interpreted with caution.

As suggested, we have more thoroughly discussed the issue of generalizability of results in the limitations of the study and have modified the title of the manuscript to take into account the peculiar characteristics of the study population. Moreover, the title has been changed in “Exploring the willingness to accept SARS-CoV-2 vaccine in a university population in Southern Italy, September to November 2020”.

A few other comments that may represent minuses of the manuscript:

the presentation of the results along the 2 models (ie in Table2), as well as their significance is relatively difficult to follow, more clarity is necessary.

As suggested, we have rephrased the presentation of the Results of Table 2 to increase clarity and significance of the results, also eliminating Odds ratio and confidence intervals that may be retrieved from Table 2. Moreover, in order the make the Table easier to follow the column regarding the standard error has been removed.

the 3rd part from Methods (referring to the behavior about influenza vaccine) was not addressed at all.

As suggested, in the Results section we have added the results related to influenza vaccine behavior.

in the Discussion, would have been good to discuss what other studies investigated the willingness to accept vaccine and maybe address the potential differences between populations

As suggested, we have added in the Discussion more studies and have underscored differences among populations.

More formal problems with the manuscript refer to necessary revision for English, duplicate entire sentence in the Abstract - surprising lack of attention (...Males, not being married...)

As suggested, we have revised the language throughout the manuscript. As regards to apparently duplicated sentences in the Abstract, the two sentences refer to two different models (concern about safety of the COVID-19 vaccine and willingness to receive the COVID-19 vaccine), in which the same variables (gender and marital status) were significant predictors. To avoid misunderstanding we have rephrased these two sentences in the Abstract.

Round 2

Reviewer 2 Report

None.

Author Response

Thank you for having reviewed the manuscript

Reviewer 3 Report

In the conclusion, specify "...considerable proportion of the studied population..." instead of "...considerable proportion of the population..."

Author Response

Thank you for having reviewed the manuscript and as suggested we have make the correction